# Intrinsic Gradient Compression for Scalable and Efficient Federated Learning

**Luke Melas-Kyriazi**[*]
Oxford University
Oxford, UK

**Franklyn Wang**[*]
Harvard University
Cambridge, MA

## Abstract

Federated learning is a rapidly growing area of research, holding the promise of privacy-preserving distributed training on edge devices. The largest barrier to wider adoption of federated learning is the communication cost of model updates, which is accentuated by the fact that many edge devices are bandwidth-constrained. At the same time, within the machine learning theory community, a separate line of research has emerged around optimizing networks within a subspace of the full space of all parameters. The dimension of the smallest subspace for which these methods still yield strong results is called the *intrinsic dimension*. In this work, we prove a general correspondence between the notions of intrinsic dimension and gradient compressibility, and we show that a family of low-bandwidth federated learning algorithms, which we call *intrinsic gradient compression algorithms*, naturally emerges from this correspondence. Finally, we conduct large-scale NLP experiments using transformer models with over 100M parameters (GPT-2 and BERT), and show that our method outperforms the state-of-the-art in gradient compression.

## 1 Introduction

Federated learning is a nascent area of study which seeks to perform machine learning in a privacy-preserving way. However, federated learning with deep neural networks suffers from a problem with communication bandwidth: it is very costly to send gradient/model updates over a network, especially when communicating with mobile phones and edge devices.

To reduce bandwidth for federated learning, it is natural to utilize various forms of compression. Previous works have tried to achieve compression in two ways: (1) by compressing the information communicated in standard gradient descent algorithms (e.g. quantizing gradients (Wen et al., 2017))

and (2) by training with non-standard methods that naturally use less bandwidth (e.g. prototypical networks (Tan et al., 2021)).

At the same time, in the machine learning theory community, researchers have been working to understand what at first seems like an entirely different question: why do hugely overparametrized models generalize so well? One promising approach to this answering this question has utilized the concept of *intrinsic dimension*, defined for a given optimization problem as the smallest dimension $d$ for which we can solve the problem when the weights are restricted to a a $d$-dimensional manifold. To be precise, it is the smallest $d$ for which the standard loss minimization problem

$$\min_{\theta' \in \mathbb{R}^d} \ell(f_{g(\theta')}) \tag{1}$$

has a satisfactory solution, where the image of $g$ is a $d$-dimensional manifold. If the intrinsic dimension of a problem is low, then even if a model is vastly overparameterized, only a small number of parameters need to be tuned in order to obtain a good solution, which is often enough to imply certain generalization guarantees.

We begin this paper by observing that the two problems above are naturally related. If one can find a solution to the problem by only tuning $d$ parameters, as in Equation (1), then a corresponding low bandwidth algorithm can be found by simply running stochastic gradient descent in the reduced parameter space (in this case, $\mathbb{R}^d$).

However, simply optimizing a subset of a model's parameters is often insufficient for training models (especially when training from scratch, rather than finetuning). Thus, we are inspired to seek a more general characterization of algorithms that use a low amount of bandwidth. In order to do this, we rewrite the optimization problem in Equation (1) in the original parameter space. When $g(\theta') = A\theta'$ for some matrix $A$ (so the low-dimensional manifold is a low-dimensional sub-

---

[*]Equal contribution

space), stochastic gradient descent can be rewritten as

$$\theta_{t+1} = \theta_t - \eta A A^\top \nabla_\theta \ell(f_\theta)|_{\theta=\theta_t}. \quad (2)$$

We call this method *static intrinsic gradient compression*, because our gradients are projected into a static ("intrinsic") subspace. Now, Equation (2) admits a natural generalization, which allows us to explore more of the parameter space while still preserving a low level of upload bandwidth usage:

$$\theta_{t+1} = \theta_t - \eta A_t A_t^\top \nabla_\theta \ell(f_\theta)|_{\theta=\theta_t} \quad (3)$$

where $A_t$ may vary with time. We call the set of all such algorithms *intrinsic gradient compression algorithms*, and consider three particular instantiations: static, time-varying, and $k$-varying, each of which perform in different use cases.

Our approach is model-agnostic and highly scalable. In experiments across multiple federated learning benchmarks (language modeling, text classification, and image classification), we vastly outperform prior gradient compression methods, and show strong performance even at very high compression rates (e.g. up to $1000\times$).

Our contributions are as follows.

- We find a general class of optimization algorithms based on the notion of intrinsic dimension that use low amounts of upload bandwidth, which we denote *intrinsic gradient compression algorithms*.

- We specify three such algorithms: static compression, time-varying compression and $K$-varying compression, with different levels of upload and download bandwidth for use in various federated settings.

- In a set of experiments, we show that these methods significantly outperform prior approaches to federated learning with gradient compression, obtaining large reductions in bandwidth at the same level of performance.

In Section 2, we describe the preliminaries needed to contextualize our work, namely ideas from intrinsic dimension, federated learning, and gradient compression. In Section 3, we show how the algorithm used by intrinsic dimension naturally generalizes to algorithms which use little upload bandwidth. In Section 4 we consider special instantiations of these algorithms in federated learning settings which attain low upload and download bandwidth, and in Section 5 show that they

achieve state of the art results. Finally, Section 6 concludes.

## 2 Preliminaries

### 2.1 Intrinsic Dimension

The concept of intrinsic dimension was introduced in the work of (Li et al., 2018), as a way of evaluating the true difficulty of an optimization problem. While this can usually be done by counting the number of parameters, some optimization problems are easier than others in that solutions may be far more plentiful.. One can write

$$\ell(f_\theta) = \ell(f_{g(\theta')}) \quad (4)$$

where $g : \mathbb{R}^d \to \mathbb{R}^D$ and thus we've transformed the problem into an optimization problem over $\theta_2$. If we can still find good solutions to the original problem where $\theta_2 \in \Theta^2$, then the problem's intrinsic dimension may be lower, and thus the question may be easier than previously expected. Throughout this paper we will always take $g(\theta') = A\theta' + \theta_0$ for a $D \times d$ matrix $A$, and take $\Theta^2 = \mathbb{R}^d$, and $\Theta^1 = \mathbb{R}^D$, where $D > d$, where $\theta_0$ is the original value of the expression.

The intrinisic dimension $g(\ell, L)$ with respect to a task $\ell$ and performance threshold $L$ is equal to the smallest integer $d$ so that optimizing Equation (4) on task $\ell$ could lead to a solution of performance at least equal to $T$. The intrinsic dimension is not exactly knowable, because we cannot find the "best performing model" exactly. However, if say, training with some optimization algorithm gives us a solution to Equation (4) with loss $\le L$ and with $d$ dimensions, we can say with certainty that $g(\ell, T) \le d$.

### 2.2 Federated Learning

Federated learning is a paradigm built around protecting the privacy of user data. The standard model involves a server and many clients, where the raw data must remain on the client's device but the server learns a model. Generally, this is implemented by only the gradients of the model on the data being sent to the central server, which then runs a standard algorithm. A common example of this is the FedAvg algorithm (McMahan et al., 2017), where models are trained to near-completion on a each client's data, and the data is then averaged. In what follows, we define an *epoch* to be a single pass over every client.

## 2.3 Gradient Compression

Sending full gradients in standard uncompressed form uses far more bandwidth than we are afforded in certain settings. For example, in a 1 billion parameter model (hardly particularly large by current standards) one gradient update would take 4 gigabytes of bandwidth uncompressed. Thus, there has been substantial amounts of work in compressing the gradient, like (Albasyoni et al., 2020), which finds an optimal gradient compression algorithm, albeit one which is computationally infeasible.

## 2.4 Related Work: Model Pruning and Model Compression

**Related Work: Model Pruning**  There has been great interest in compressing models by using fewer weights, starting with the work of (Hinton et al., 2015; Han et al., 2015). One related work is *Diff Pruning* (Guo et al., 2021), which constrains the number of weights that can be changed from a pretrained model. In essence, diff pruning attempts to solve an $L^0$ minimization problem on the weights of the model, and approaches this by means of a relaxation.

A number of other works have explored the idea of finetuning by only modifying a subset of a model's parameters. (Jiang et al., 2019) and (Bibikar et al., 2021) utilize sparsity to reduce communication costs during training. (Ravfogel et al., 2021) finetunes only the layer biases of large models. Similarly, (Houlsby et al., 2019) finetunes low-parameter adapters between each layer. Compared to (Ravfogel et al., 2021) our method is far more flexible, allowing any number of parameters to be changed. Compared to (Houlsby et al., 2019) our methods are architecture-independent, and can be applied to any model.

**Related Work: Federated Learning**  Federated learning is a machine learning paradigm in which a model is trained by a collection of clients, each with their own private local data. From the introduction of federated learning (McMahan et al., 2017), it was clear that communication costs represented a significant challenge: sending gradients or weights over a network is costly due to the large size of modern machine learning models. (McMahan et al., 2017) introduced the FedAvg algorithm, which aims to reduce communication costs by sending and averaging weights, rather than gradients. Specifically, clients train their model locally for a given number of epochs, send it to the server, and received an averaged copy of the model weights. However, sending the full set of model weights often remains very costly (especially when clients only have a small amount of local data, such that many rounds of communication are necessary); as a result, FedAvg performs poorly in heavily-bandwidth-constrained settings.

Recently, FetchSGD (Rothchild et al., 2020) aimed to address this issue differently by utilizing the concept of sketching. Rather than transmitting full gradients from the client to the server, they send a sketch of the gradient. This approach performs well, but only yields moderate compression rates. We compare to FetchSGD in Section 5.

# 3 A Family of Low-Bandwidth Algorithms

In this section, we characterize a family of low-bandwidth optimization algorithms based on the notion of intrinsic dimension.

We start from the optimization problem induced by intrinsic dimension (Equation (4)). If we directly run gradient descent on Equation (4) with respect to the intrinsic weights $\theta'$, we obtain an equation of the following form:

$$\theta'_{t+1} = \theta'_t - \eta \nabla_{\theta'} \left( \ell(f_{g(\theta')}) \right) = \theta'_t - \eta \nabla_{\theta'} \left( \ell(f_{A\theta'}) \right)$$
$$= \theta'_t - \eta A^\top \nabla_\theta (\ell(f_\theta))^\top |_{\theta = A\theta'_t + \theta_0}$$

Then, left-multiplying both sides by $A$ we obtain

$$\theta_{t+1} = \theta_t - \eta \, A \underbrace{\underbrace{A^\top \nabla_\theta(\ell(f_\theta))|_{\theta=\theta_t}}_{\text{compressed gradient}}}_{\text{approximate gradient}} \quad (5)$$

Note that here, we can interpret $A^\top \nabla_\theta(\ell(f(\theta)))|_{\theta=\theta_t}$ as a compressed gradient with dimension $d$, and $AA^\top \nabla_\theta(\ell(f(\theta)))|_{\theta=\theta_t}$ as the approximate gradient. This inspires us to consider the more general family of optimization algorithms given by

$$\theta_{t+1} = \theta_t - \eta A_t A_t^\top(\boldsymbol{v}_t), \quad (6)$$

where $\boldsymbol{v}_t$ is a $D$ dimensional vector computed from data available at timestep $t$ that plays a similar role to a gradient, but may not be an exact gradient, and the $A_t$ are all $D \times d$ matrices known ahead of time (say, generated with random seeds). One intuitive way of interpreting this algorithm is that $\theta_{t+1} - \theta_t$ is constrained to lie in a low-dimensional subspace,

**Algorithm 1** Static Intrinsic Gradient Compression

---
**input:** learning rate $\eta$, timesteps $T$, local batch size $\ell$, clients per round $W$
Create matrix $A \in \mathbb{R}^{D \times d}$ with $\mathbb{E}[AA^\top] = I_D$.
Current Vector: $\Sigma_0 = 0$
**for** $t = 1, 2 \cdots T$ **do**
    Randomly select $W$ clients $c_1, \ldots c_W$.
    **loop**
        {In parallel on clients $\{c_i\}_{i=1}^W$}
        Download $\Sigma_{t-1}$, calculate current $\theta_{t-1} = \theta_0 + A(\Sigma_{t-1})$.
        Compute stochastic gradient $g_i^t$ on batch $B_i$ of size $\ell$:
        $g_i^t = \frac{1}{\ell} \sum_{j=1}^\ell \nabla_\theta \mathcal{L}(\theta_{t-1}, z_j)$.
        Sketch $g_i^t$ to $S_i^t = A^\top g_i^t$ and upload it to the aggregator.
    **end loop**
    Aggregate sketches $S^t = \frac{1}{W} \sum_{i=1}^W S_i^t$
    Unsketch: $\Delta_t = A S^t$
    Update: $\theta_t = \theta_{t-1} - \eta \Delta_t$, $\Sigma_t = \Sigma_{t-1} - \eta S^t$.
**end for**

---

namely that given by the span of $A_t$. This family of algorithms can be made to use only $d$ upload bandwidth, as only the vector $A_t^\top(\boldsymbol{v}_t)$ must be uploaded. Furthermore, note that Equation (6) has no references to the intrinsic weights $\theta'$, meaning that it represents a general optimization algorithm in the original space. Formally,

**Theorem 3.1.** *All algorithms of the form*

$$\theta_{t+1} = \theta_t - \eta A_t A_t^\top(\boldsymbol{v}_t)$$

*can be simulated with $d$ upload bandwidth in a standard federated learning setting, where $\boldsymbol{v}_t$ is a function that can be calculated by the client at time $t$ combined with all data from the server.*

We call all such algorithms *intrinsic gradient compression algorithms*. Note that this theorem only bounds the upload bandwidth capacity needed to run gradient descent, and does not bound the download bandwidth. In the particular instantiations we consider, we will demonstrate that one can also bound the download bandwidth.

## 4 Intrinsic Gradient Compression Algorithms

While Theorem 3.1 shows that any algorithm of the form Equation (6) can be implemented with low levels of upload bandwidth, not every algorithm of the form Equation (6) can be implemented with low levels of download bandwidth as well. Theorem 3.1 gives rise to a family of algorithms we denote *intrinsic gradient compression algorithms*. In this section, we describe three particular intrinsic gradient compression algorithms which use low amounts of both upload and download bandwidth.

These federated learning algorithms can be decomposed into three main phases.

- **Reconciliation:** The client reconciles its model with the server's copy of the model.

- **Compression:** The local model calculates, compresses, and sends its local gradient to the server.

- **Decompression:** The server model updates its own copy of the model using the estimated gradient from the local model.

In general, reconciliation will be by far the most complex part of each algorithm, and the other steps are essentially shared across algorithms.

We show how to implement SGD for each variant, and note that this choice of optimization algorithm is quite necessary – other optimization algorithms like SGD with momentum cause the parameters to not move in the low-dimensional subspace, which makes the compression impossible. While one can implement a variant which resets the momentum every epoch, momentum is rarely a useful optimization in federated learning due to the non-i.i.d. nature of the batches) so we do not consider this.

**Static Intrinsic Gradient Compression** In this subsection, we seek to implement the static intrinsic gradient compression algorithm

$$\theta_t = \theta_{t-1} - \eta A A^\top \nabla_\theta \mathcal{L}(\theta_{t-1})$$

in a federated learning setting.

In the reconciliation phase, since we know that the parameters $\theta^c$ (which denotes the current parameters of the server) will always be equal to $\theta_0 + A\Sigma$ for some $\Sigma \in \mathbb{R}^d$, the server can just send $\Sigma$ to the client, which will take $d$ download bandwidth.

For compression, the client compresses the gradient by multiplying by $A^\top$, and for decompression the server multiplies this by $A$. The full algorithm is given in Algorithm 1.

**Time-Varying Intrinsic Gradient Compression** In this subsection, we implement the time-varying intrinsic gradient compression algorithm

$$\theta_t = \theta_{t-1} - \eta A_e A_e^\top \nabla_\theta \mathcal{L}(\theta_{t-1})$$

in a federated learning setting, where $e$ is the epoch.

In this case, we show that our algorithm can be implemented with at most $2d$ bandwidth used per

| Intrinsic Gradient Compression Method | Upload | Download | Dimensions Explored |
|---|---|---|---|
| Static | $dE$ | $dE$ | $d$ |
| Time-Varying | $dE$ | $2dE$ | $dE$ |
| $K$-Varying | $dE$ | $2dEK$ | $dEK$ |
| No Compression | $DE$ | $DE$ | $D$ |

Table 1: Bandwidth and Performance Comparisons. The bandwidth refers to that of that used for each client. Note that we break upload and download bandwidth into separate columns, because download speeds can often be considerably faster than upload speeds and we may thus be willing to tolerate higher values of download bandwidth. A realistic example of the values of the variables above is e.g. $d = 10^3, D = 10^8, E = 20, K = 8$.

client per timestep, so over $E$ epochs there is $2dE$ bandwidth used total on downloading. Since this bandwidth is twice that of static subspace compression, but we search $E$ times more directions in the space, this algorithm is particularly useful when we have many epochs.

Letting $\theta_e^c$ be the client parameters at epoch $e$, note that we have the value of $\theta_{e-1}^c$ when performing reconciliation. Now we can write

$$\theta_e^c - \theta_{e-1}^c = (\theta_e^c - \theta_{e-1}^{\text{final}}) + (\theta_{e-1}^{\text{final}} - \theta_{e-1}^c)$$

and we can see that $(\theta_e^c - \theta_{e-1}^{\text{final}})$ lies in the column space of $A_e$ and $(\theta_{e-1}^{\text{final}} - \theta_{e-1}^c)$ lies in the column space of $A_{e-1}$, which is enough to find the full algorithm, given in Algorithm 2.

$K$-**Varying Intrinsic Gradient Compression**   In this subsection, we describe how to implement the $K$-varying intrinsic gradient compression algorithm

$$\theta_t = \theta_{t-1} - \eta A_e^{(i)} A_e^{(i)\top} \nabla_\theta \mathcal{L}(\theta_{t-1})$$

where $\{A_e^{(i)}\}_{i=1}^K$ is the set of $K$ compression matrices used at epoch $e$, and $i$ is a randomly chosen integer between 1 and $K$ inclusive.

This method is motivated from the fact that in many cases, the upload speed is much slower than the download speed, so we may only want to project the gradient into part of the subspace currently being explored, as opposed to the complete subspace. This allows each client to explore $d$ directions at a time, but for $dK$ directions to be explored across the entire epoch. As such, the algorithm identical time-varying compression, and is given in Algorithm 3.

**Choice of Compression Matrix**   Finally, we we discuss the choice of compression matrix for $A$. We note that our methods are agnostic to the specific

choice of $A$, and depend only on the existence of efficient subroutines for calculating the matrix-vector products $Ax$ and $A^\top y$. Nonetheless, the choice of $A$ has significant implications for the resulting accuracy of the algorithms. In order to maintain the most proximity to the original stochastic gradient descent algorithm, we will choose normalized $A$ so that $\mathbb{E}[AA^\top] = I_D$.

The naive choice is to let $A$ be a $D \times d$ random dense matrix, but such a choice is impossible due to memory constraints. For example, if we aim to train even a small version of BERT (100M parameters) with an intrinsic dimension of 1000, we would need to store a matrix with $10^{11}$ entries.

The approach taken by (Aghajanyan et al., 2021; Li et al., 2018) for large-scale experiments, which we follow, utilizes the *Fastfood transform* (Le et al., 2013), in which $A$ can be expressed as the $D \times d$ matrix $A_i = \text{Unpad}_D B_i H \Pi_i G_i H \text{Pad}_{2^\ell}$ where $2^\ell$ is the smallest power of two larger than $D$, $H$ is a standard Hadamard matrix, $B_i$ is a random diagonal matrix with independent Rademacher entries (random signs), $\Pi$ is a random permutation matrix, $G$ is a random diagonal matrix with independent standard normal entries, $\text{Pad}_{2^\ell}$ to be a linear operator which simply pads a $d$-dimensional vector $v$ with zeroes until it has size $2^\ell$, and $\text{Unpad}_D$ is a linear operator which takes the first $D$ elements from a $2^\ell$-dimensional vector. Since we can quickly compute a matrix-vector product by $H$ with a fast Walsh-Hadamard transform, we can perform a matrix multiplication by $A_i A_i^\top$ in $O(\ell 2^\ell) = O(D \log D)$ time and $O(D)$ space.

**Performance Comparison**   We show the theoretical tradeoffs between each of these algorithms in Table 1.

| | Name | Intrinsic Dim. | PPL | Up. Comp. | Down. Comp. | Total Comp. |
|---|---|---|---|---|---|---|
| | Uncompressed | | 13.9 | 1 | 1 | 1 |
| (McMahan et al., 2017) | FedAvg (2 local iters) | | 16.3 | 2 | 2 | 2 |
| (McMahan et al., 2017) | FedAvg (5 local iters) | | 20.1 | 5 | 5 | 5 |
| | Local Top-K ($k = 50,000$) | | 19.3 | 30.3 | 2490 | 60 |
| | Local Top-K ($k = 500,000$) | | 17.1 | 3.6 | 248 | 7.1 |
| (Rothchild et al., 2020) | FetchSGD ($k = 25,000$) | | **14.8** | 3.8 | 100 | 7.3 |
| (Rothchild et al., 2020) | FetchSGD ($k = 50,000$) | | 15.8 | 2.4 | 10 | 3.9 |
| | Ours (static) | 16384 | 27.7 | 7595 | 7595 | 7595 |
| | Ours ($K$-subspace) | 16384 | 19.6 | 7595 | 949 | 1688 |
| | Ours (static) | 65536 | 20.6 | 1900 | 1900 | 1900 |
| | Ours ($K$-subspace) | 65536 | 17.8 | 1900 | 237 | 422 |
| | Ours (static) | 262144 | 17.6 | 475 | 475 | 475 |
| | Ours ($K$-subspace) | 262144 | 16.6 | 475 | 59.3 | 105 |
| | Ours (static) | 1048576 | 15.8 | 119 | 119 | 119 |
| | Ours ($K$-subspace) | 1048576 | 15.4 | 119 | 14.8 | 26.3 |
| | Ours (static) | 4194304 | **14.8** | 29.7 | 29.7 | 29.7 |

Table 2: Language modeling perplexity (lower is better) and compression rates (higher is better) for a GPT-2 model (124M parameters) on the PersonaChat dataset. We compare to prior work, including the state-of-the-art in gradient compression (FetchSGD), and we show upload, download, and total compression rates. For our intrinsic gradient compression results, we give static and $K$-subspace compression for a range of dimensions between 16386 and 4194304. For $K$-subspace compression we use $K = 8$. Overall, we match or exceed the performance of prior work with significantly improved compression rates.

## 5   Experiments

We evaluate our method across a range of benchmarks to showcase the potential of our three algorithms. These include two natural language processing tasks (language modeling and text classification), as well as a computer vision task (image classification).

As with previous works (Rothchild et al., 2020; McMahan et al., 2017), we simulate the federated learning in order to scale to large numbers of clients (upwards of 10,000). We simulate on 8 commercial-grade GPUs for the language modeling experiments and 1 GPU for the other experiments. We perform experiments in both non-IID (language modeling, image classification) and IID (text classification) settings, because both scenarios are common in real-world federated learning.

**Image Classification (ResNet-9 on CIFAR-10)**
First, we consider image classification on the CIFAR-10 dataset, a collection of 50,000 images with resolution $32 \times 32$px. We use the same experimental setup as (Rothchild et al., 2020): we split the data between 10,000 clients in a non-IID fashion, such that each client only has data from a single class. At each step, we sample 100 clients at random, such that each gradient step corresponds to 500 images. We perform 24 rounds of communi-

cation between all clients (i.e. 24 training epochs).

We use a ResNet-9 architecture with 6,570,880 trainable parameters for our fair comparison to previous work. Note that the model does not have batch normalization, as batch normalization would not make sense in a setting where each client has so few examples. Due to the substantial number of epochs performed here, we experiment with both static and time-varying gradient compression ($k$-varying compression is better suited to settings involving fewer rounds of communication). We perform experiments across intrinsic dimensions of 4000, 8000, 16000, 32000, 64000, 128000, and 256000.

Our results are shown in Figure 1. Whereas FedAvg and Top-K struggle at even modest compression rates (e.g. $3\times$), the intrinsic gradient compression methods deliver strong performance at much larger compression rates. The intrinsic methods outperform the current state-of-the-art gradient compression method, FetchSGD (Rothchild et al., 2020), by a large margin, and easily scale to high compression rates (e.g. $100\times$). Finally, we see that time-varying intrinsic compression generally outperforms static compression for the same communication cost.

**Text Classification (BERT on SST-2)**   Next, we consider text classification on the Stanford Senti-

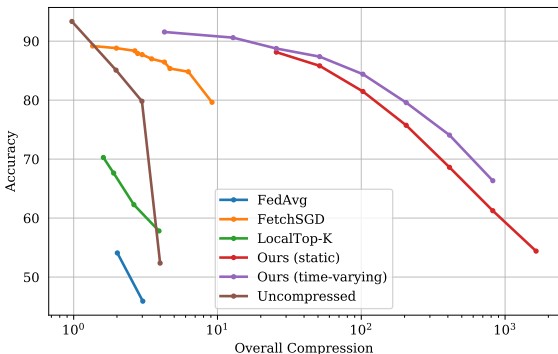

(a) Final Accuracies on CIFAR-10 with differing levels of compression.

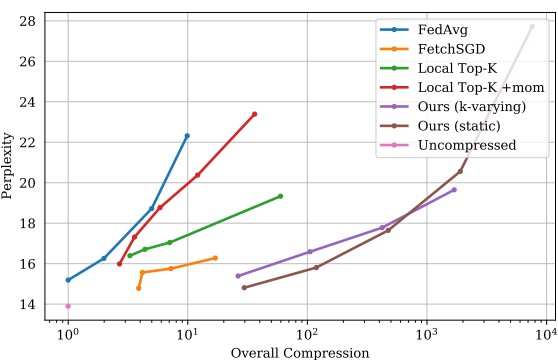

(a) Perplexity on PersonaChat compared to other recent federated learning methods

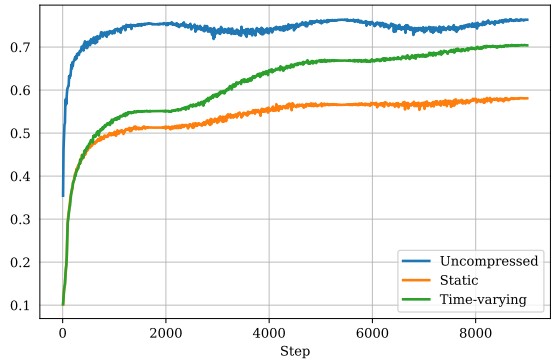

(b) Training curves on CIFAR-10 with static and time varying dimension at the same intrinsic dimensionality.

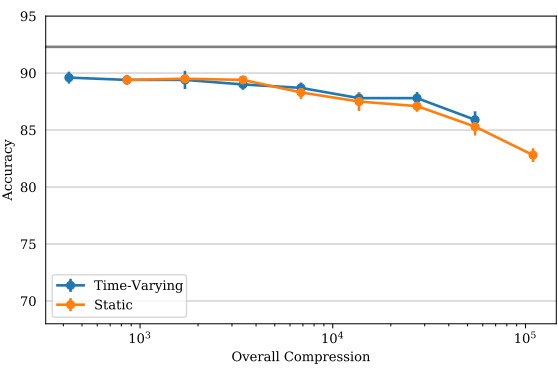

(b) Accuracy on SST-2

Figure 1: Results on computer vision benchmarks. Both static and time-varying intrinsic gradient dimension significantly outperform perform work, with time-varying intrinsic compression performing best. On the right, we see that time-varying and static compression perform similarly at the beginning of training, but time-varying outperforms static eventually but are tied at the beginning, and that time-varying outperforms static with equal space. For the FedAvg and uncompressed methods with compression rates above 1, compression was performed by training for fewer epochs.

ment Treebank-v2 (SST-2) dataset (Socher et al., 2013), a common sentiment analysis dataset. For this experiment, we consider an IID data split into 50 and 500 clients, respectively. We employ the popular BERT (Devlin et al., 2019) transformer architecture with 109M parameters. The purpose of this experiment is to push the limits of gradient compression; we project the 109M-dimension BERT gradients into as few as 200 dimensions.

We perform 30 rounds (i.e. 30 epochs) of training for all compressed runs, while we perform 6 for the uncompressed baseline (as it converges more quickly). Federated learning experiments has previously been criticized for being challenging to reproduce; as a result, we perform each run five

Figure 2: Results on NLP benchmarks. Note that while $K$-varying appears to perform poorly on PersonaChat, the upload performance is much stronger. See Appendix D for these full results.

times over different random seeds. We report the mean, min, max, and standard deviation of the runs in Appendix D.

Due to the substantial number of epochs performed here, it is natural to apply static and time-varying intrinsic gradient compression. We use intrinsic dimensions of 200, 400, 800, . . . , 25600.

Our results are given in Figure 2. First, along similar lines to (Aghajanyan et al., 2021), we find that it is possible to achieve remarkably high compression ratios for text classification: we achieve close to full performance even when compressing the 109M-dimension parameter vector into an intrinsic space of dimension 16,384. Furthermore, we find that time-varying intrinsic gradient compression consistently outperforms static intrinsic gradient compression at the same compression rate.

**Language Modeling (GPT-2 on PersonaChat)**
Lastly, we consider language modeling on the PersonaChat (Zhang et al., 2018) dataset of dialogues between Amazon Mechanical Turk workers as-

signed to act out specific personalities. [1] The dataset has a non-IID split into 17,568 clients in which each client is assigned all data corresponding to given personality; as a result, it is widely used in federated learning simulations. We perform language modeling using the GPT-2 transformer architecture (124M parameters). For fair comparison to previous work, we conduct only two rounds of training across the clients (i.e. two epochs).

Due to the low number of training rounds, it is natural to apply *static* and *K-varying* gradient compression.[2] Specifically, we apply both of these algorithms to train GPT-2 using intrinsic dimensions of 16384, 65536, 262144, 1048576, and 4194304.

Our results are shown in Figure 2. Overall, intrinsic dimension-based gradient compression vastly outperforms a wide range of prior approaches to reducing communication in federated learning. On the low-compression end of the spectrum, we obtain nearly full performance with superior compression rates to FedAvg (McMahan et al., 2017) and the recent FetchSGD (Rothchild et al., 2020). On the high-compression end of the spectrum, we scale better than previous approaches. For example, we obtain a perplexity of around 20 even with an extremely high compression rate of 1898.

Finally, we see that $K$-varying intrinsic compression performs similarly to (or slightly worse) than static compression at the same level of overall compression. However, if it is more important to conserve upload bandwidth than download bandwidth, then $K$-varying intrinsic gradient compression significantly outperforms static intrinsic gradient compression (see Section 4).

### 5.1 Gradient Compression Results

One of the primary motivations of federated learning is the desire for individual clients to be able to retain data privacy while still participating in model training.

However, a number of works have shown that if the client does not have a large amount of data

and the client sends back their full local gradient, it is possible to approximately reconstruct their local data from the model. This is a significant problem, because their data would then effectively be visible to the central server and any attackers that intercept their communications.

Here, we show that compressing gradients with our approach can mitigate this problem. Specifically, we check if our compressed gradients can be reconstructed with the procedure proposed by (Zhu et al., 2019). As in (Zhu et al., 2019), we use a ResNet-152 model a randomly selected image from ImageNet and run for 24,000 iterations (by which time the method has converged). We reconstruct the image both from the full gradient (the center image) and from a the intrinsically-compressed image (the right image) with intrinsic dimension 65,536.

As seen in Figure 3, given the full gradient it is possible to obtain a fairly good reconstruction of the image. By contrast, with our method, the reconstruction is visually much less similar from original image. Of course, our method does not solve the problem entirely; an outline of the dog in the image is still visible because the compressed gradient still contains some information about the local data. To solve the issue entirely, it would be necessary to use a method such as differential privacy.

## 6 Conclusion

Federated learning holds the promise of large-scale model training while simultaneously letting users retain control over their data. In this paper, we preset a set of novel algorithms for scalable and efficient federated learning. These algorithms are particularly helpful for NLP training, where models often have hundreds of millions of parameters. Our experiments finetuning BERT and GPT-2 that our proposed method significantly improves upon the state-of-the-art in gradient compression for federated learning. In future work, we hope to deploy our system in a real-world federated learning setting with a large number of physical devices, rather than solely in simulation.

## 7 Acknowledgments

We would like to thank Ankur Moitra, Yang Liu, and Demi Guo for helpful discussions. L. M. K. is supported by the Rhodes Trust.

---

[1] In more detail, the PersonaChat dataset (Zhang et al., 2018) was collected by first giving imaginary personas (defined by a set of 5 sentences) to Amazon Mechanical Turk workers and asking them to take on those personas. Then, the system paired workers and asked them to discuss. Since the personas were imaginary and no personally identifiable information was exchanged (in particular, the workers were explicitly told to not use personally identifiable information) the dataset does not contain personally identifiable information.

[2] Time-varying compression does not make sense here, as its benefit is derived from the setting where there are many rounds of communication between the clients.

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

# Appendix

## A  Algorithms

In Algorithm 2 and Algorithm 3 below, we provide the full time-varying and $K$-varying intrinsic gradient compression algorithms, which were omitted from the main text.

## B  Proofs

### B.1  Proof of Theorem 3.1

First, note that the server knows the value of $A_t$. Then, for any local vector $v_t$, the client can send $A_t^\top(v_t)$ to the server, and the server can calculate $A_t A_t^\top$, enabling it to continue executing the algorithm.

## C  Additional Related work

In the main paper, we described the prior work in federated learning and machine learning theory that was directly relevant to our paper's method. Here, we describe a number of less-related works that could not be included in the main paper due to space constraints.

**Intrinsic Dimensionality**   As mentioned in the main paper, the concept of measuring the intrinsic dimensional of loss landscapes was introduced by (Li et al., 2018). (Li et al., 2018) consider optimizing a $D$-parameter model in a random $d$-dimensional subspace of the full parameter space. They define the intrinsic dimension of the optimization problem as the minimum dimension $d$ for which a solution to the problem can be found, where a "solution" refers attaining a certain percentage of the maximum possible validation accuracy (i.e. the validation accuracy obtained by optimizing in all $D$ dimensions). They use a fixed cut-off of 90% accuracy for their experiments.

(Aghajanyan et al., 2021) followed up on this work by considering the setting of finetuning models in natural language processing. They show that the intrinsic dimension of some of these tasks (e.g. text classification on MRPC) is surprisingly low.

A number of works have tried to measure the intrinsic dimension of datasets, rather than objective landscapes. (Levina and Bickel, 2005) introduced a maximum likelihood approach to estimating intrinsic dimensionality based on nearest-neighbors, while (Ceruti et al., 2014) employed angle and norm-based similarity. More recently, () further

extended this line of work to use minimal neighborhood information.

Finally, some works have tried to measure the intrinsic dimensionality of image representations and datasets. (Gong et al., 2019) finds that the representations produced by popular image and face representation learning models (ResNet-50 and SphereFace) have quite low intrinsic dimensionalities (16 and 19, respectively). Along similar lines, (Pope et al., 2021) showed that popular image datasets (MNIST, CIFAR 10, ImageNet) also have low intrinsic dimensionality.

**Federated Learning**   Federated learning is generally concerned with the distributed training of machine learning models across many devices, each of which holds private data. Many aspects of this federated setup are separate subfields of research, including how to ensure the privacy of client-held data (Xie et al., 2020; Bhagoji et al., 2019), how to deal with heterogeneous data and networks (Li et al., 2020a,b; Yu et al., 2020), how to reconcile weights/gradients from multiple clients (Li et al., 2020a; Wang et al., 2020; Li et al., 2020c), how to manage clients in a fault-tolerant manner, deployment on mobile/iot devices (He et al., 2020), and fairness (Mohri et al., 2019).

Numerous works focus on making federated training more efficient, with the ultimate goal of reducing communication cost and training time. The classic FedAvg (McMahan et al., 2017) algorithm tries to do this by communicating weights rather than gradients. FedProx (Li et al., 2020a) generalizes and re-parametrizes FedAvg. FedMA (Wang et al., 2020) continues to improve this approach by matching and averaging hidden layers of networks with similar activations at each communication round. FedAwS (Yu et al., 2020) considers federated averaging in the case where each client has data from only a single class. (Malinovsky et al., 2020) analyzes a generalization of these weight-averaging approaches from a theoretical viewpoint.

Relative to the weight averaging approach, the approach of compressing and sending gradients is relatively understudied. (Albasyoni et al., 2020) describes an approach that is theoretically optimal but not practical for large non-linear models. (Han et al., 2020) proposes adaptive gradient sparsification for federated learning, in which a subset of the full gradient is communicated at each round. FetchSGD (Rothchild et al., 2020) compresses gradients by sketching; it is the current state-of-the-art

---

**Algorithm 2** Time-Varying Intrinsic Gradient Compression

---

**input:** learning rate $\eta$, timesteps $T$, local batch size $\ell$, clients per round $W$
**for** $e = 1, 2, \cdots E$ **do**
    Create matrix $A_e \overset{\text{i.i.d.}}{\sim} A$ where $A \in \mathbb{R}^{D \times d}$ with $\mathbb{E}[AA^\top] = I_D$.
    Current, Final Vector: $\Sigma_e^{\text{current}} = 0, \Sigma_e^{\text{final}} = 0$
    **for** $t = 1, 2 \cdots T$ **do**
      Randomly select $W$ clients $c_1, \ldots c_W$.
      **loop**
        {In parallel on clients $\{c_i\}_{i=1}^W$}
        Download $\Sigma_e^{\text{current}}, \Sigma_{e-1}^{\text{final}}$, calculate current $\theta_e^{c_i} = \theta_{e-1}^{c_i} + A_{e-1}(\Sigma_{e-1}^{\text{final}} - \Sigma^{\text{last}}) + A_e(\Sigma_e^{\text{current}})$.
        Update $\Sigma^{\text{last}} = \Sigma_e^{\text{current}}$.
        Compute stochastic gradient $g_i^t$ on batch $B_i$ of size $\ell$: $g_i^t = \frac{1}{\ell} \sum_{j=1}^\ell \nabla_\theta \mathcal{L}(\theta_e^{c_i}, z_j)$.
        Sketch $g_i^t$ : $S_i^{(e)t} = A_e^\top g_i^t$ and upload it to the aggregator.
      **end loop**
      Aggregate sketches $S^{(e)t} = \frac{1}{W} \sum_{i=1}^W S_i^{(e)t}$
      Unsketch: $\Delta^{(e)t} = A_e S^{(e)t}$
      Update: $\theta^{\text{current}} = \theta^{\text{current}} - \eta \Delta^{(e)t}, \Sigma_e^{\text{current}} = \Sigma_e^{\text{current}} - \eta S^{(e)t}$.
    **end for**
    Let $\Sigma_e^{\text{final}} = \Sigma_e^{\text{current}}$.
**end for**

---

in gradient compression for federated learning. We describe it in further depth in the main paper.

Finally, (Reddi et al., 2021) and (Li et al., 2020c) accelerate training by bringing adaptive optimizers built for centralized learning into the federated setting.

## D   Further Experimental Analysis

In the main paper, we included a number of figures demonstrating our performance in comparison to prior work. Here, we include tables with our precise results for clarity and in order to facilitate future comparison with our work.

### D.1   Further PersonaChat Analysis

Section 4 shows full results on PersonaChat, complete with upload and download compression. Overall compression is calculated as average compression over both upload and download.

We compare with FedAvg (McMahan et al., 2017), Top-K, and FetchSGD (Rothchild et al., 2020). FedAvg is the baseline federated learning approach involving sending and averaging weights. Top-K refers to sending the top gradients, sorted by magnitude. FetchSGD compresses the weights with sketching.

Our method significantly outperforms competing approaches across the board. We obtain an accuracy close to that of uncompressed optimization using INSERTx overall compression; FedAvg and Top-K both fail to achieve such strong results, while FetchSGD does so at a significantly lower compression rate.

Next we compare static and K-varying intrinsic gradient compression. When comparing overall compression rates, static compression is slightly better than K-varying compression. However, K-varying compression is optimized for low upload bandwidth; it obtains much better upload compression rates than static compression at the same accuracy. For example, K-varying compression with $k = 8$ and $d = 65536$ yields perplexity 17.6 at upload compression $1900\times$, whereas static compression with $d = 262144$ yields perplexity 17.4 at upload compression $475\times$.

### D.2   Further SST-2 Analysis

In Table 3, we show full results for the SST-2 dataset with static and time-varying gradient compression for a range of intrinsic dimensions. We include in this experiment an demonstration of the robustness of our method to variation in random seeds; we run each experiment five times using separate random seeds (i.e. different intrinsic subspaces and model initializations). We report standard errors in Table 3; variability is quite low.

We also see that time-varying intrinsic gradient compression outperforms static intrinsic compression, especially for low intrinsic dimensions. For example, time-varying compression at $d = 200$ outperforms static compression with $d = 400$, and time-varying compression with $d = 400$ outperforms static compression with $d = 800$.

---

**Algorithm 3** $K$-Varying Intrinsic Gradient Compression

---

**input:** distinct subspaces $K$, learning rate $\eta$, timesteps $T$, local batch size $\ell$, clients per round $W$

**for** $e = 1, 2, \ldots E$ **do**

    Create matrices $A_e^{(1)}, A_e^{(2)}, \ldots A_e^{(K)} \overset{\text{i.i.d.}}{\sim} A$ where $A \in \mathbb{R}^{D \times d}$ with $\mathbb{E}[AA^\top] = I_D$.

    Current, Final Vector: $\Sigma_e^{\text{current}(k)} = 0$, $\Sigma_e^{\text{final}(k)} = 0$ for $k = 1, 2, \ldots K$.

    **for** $t = 1, 2 \cdots T$ **do**

        Randomly select $W$ clients $c_1, \ldots c_W$.

        **loop**

            {In parallel on clients $\{c_i\}_{i=1}^W$}

            Download $\Sigma_e^{\text{current}(k)}, \Sigma_{e-1}^{\text{final}(k)}$ for $k = 1, \ldots K$, and calculate:

            $\theta_e^{c_i} = \theta_{e-1}^{c_i} + \sum_{k=1}^K \left( A_{e-1}(\Sigma_{e-1}^{\text{final}(k)} - \Sigma^{\text{last}(k)}) + A_e(\Sigma_e^{\text{current}(k)}). \right)$

            $\Sigma^{\text{last}(k)} = \Sigma_e^{c(k)}$ for $k = 1, 2, \ldots K$.

            Choose a random $k_1 \sim \text{DUnif}(\{1, 2, \ldots K\})$

            Compute stochastic gradient $g_i^t$ on batch $B_i$ of size $\ell$: $g_i^t = \frac{1}{\ell} \sum_{j=1}^\ell \nabla_\theta \mathcal{L}(\theta_e^{c_i}, z_j)$.

            Sketch $g_i^t : S_i^{(e)t} = (k_1, A_e^{(k_1)\top} g_i^t)$ and upload it to the aggregator.

        **end loop**

        Write sketches received as $\{S_w^{(e)t}\}_{w=1}^W = \{(j_w, C_w^{(e)t})\}_{w=1}^W$.

        Unsketch $S^{(e)t}$ to get $\Delta^{(e)t} = \frac{1}{W} \sum_{w=1}^W A_e^{(j_w)} C_w^{(e)t}$

        Update: $\theta^{\text{current}} = \theta^{\text{current}} - \eta \Delta^{(e)t}$,

        **for** $k = 1, 2 \ldots K$ **do**

            Update: $\Sigma_e^{\text{current}(k)} = \Sigma_e^{\text{current}(k)} - \frac{\eta}{W} \sum_{w, j_w = k} C_w^{(e)t}$.

        **end for**

    **end for**

    Let $\Sigma_e^{\text{final}(k)} = \Sigma_e^{c(k)}$ for $k = 1, 2, \ldots K$.

**end for**

---

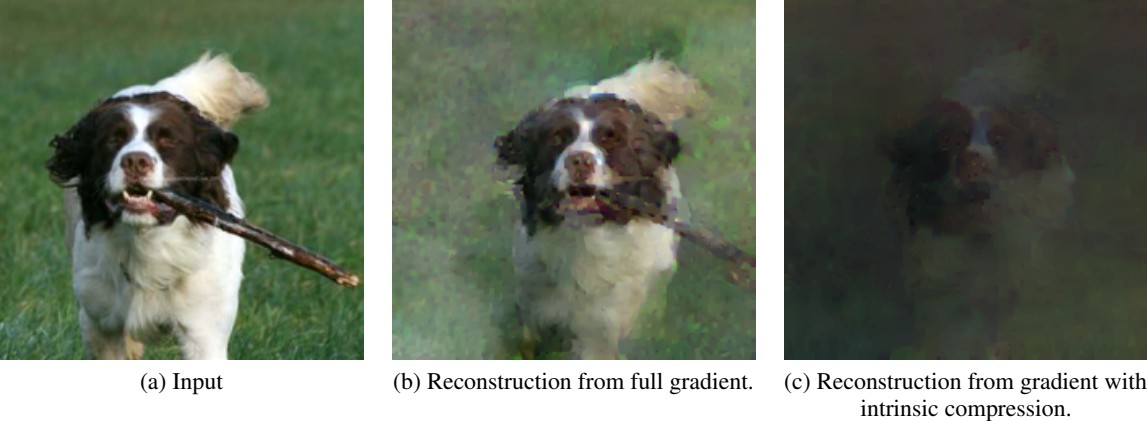

(a) Input

(b) Reconstruction from full gradient.

(c) Reconstruction from gradient with intrinsic compression.

Figure 3: Image reconstruction from gradients with and without our intrinsic gradient compression method. On the left, we show the original image. In the center, we show the result of reconstructing the image from a single gradient from a ResNet-152 model (60M parameters), produced using the method of (Zhu et al., 2019). On the right, we show the result of the same image reconstruction method applied to an gradient compressed by our algorithm using intrinsic dimension 65,536.

| Intrinsic Dim. | 200 | 400 | 800 | 1,600 |
|---|---|---|---|---|
| Static | 82.8 ($\pm$0.69) | 85.3 ($\pm$0.89) | 87.1 ($\pm$0.57) | 87.5 ($\pm$0.94) |
| Time-Varying | 85.9 ($\pm$0.85) | 87.8 ($\pm$0.61) | 87.8 ($\pm$0.59) | 88.7 ($\pm$0.54) |

| Intrinsic Dim. | 3,200 | 6,400 | 12,800 | 25,600 |
|---|---|---|---|---|
| Static | 88.3 ($\pm$0.65) | 89.4 ($\pm$0.33) | 89.5 ($\pm$0.21) | 89.5 ($\pm$0.21) |
| Time-Varying | 89.0 ($\pm$0.53) | 89.4 ($\pm$0.91) | 89.4 ($\pm$0.19) | 89.4 ($\pm$0.19) |

Table 3: Accuracy and standard error of a BERT model trained on the Stanford Sentiment Treebank v2 (SST-2) for varying intrinsic dimensions. We calculate the standard error over five trials with different random seeds. We see that for fixed dimension, time-varying intrinsic gradient compression outperforms static intrinsic gradient compression.