# OpenReview forum: "Intrinsic Gradient Compression for Scalable and Efficient Federated Learning"
_aclweb.org/ACL/2022/Workshop/FL4NLP — FL4NLP@ACL2022_

### Official Review · Reviewer_sQhs · 2022-03-21
**The authors propose a set of communication-efficient federated learning algorithms that are based on the prior idea of intrinsic dimension in theoretical machine learning.**

**Rating:** 7
**Confidence:** 4

**Review:**

The authors propose a set of communication-efficient federated learning algorithms that are based on the prior idea of intrinsic dimension in theoretical machine learning. Essentially, it has been known in theoretical ML that in the overall parameter space of the ML model, there is an intrinsic subspace, with potentially much smaller dimension than the model parameter space, where optimization can be carried out. Exploiting this concept and related ideas on intrinsic dimension, the authors propose a set of three novel strategies that enables compression of updates communicated between the FL server and clients, reducing the communication load dramatically. The underlying idea is to use a projection matrix for compression at the clients/decompression at the server, so that both training and global update can be done in the model parameter space while communications can be done in the lower dimensional space. The first algorithm considers the projection matrix to be fixed throughout training, the other two consider different versions of variable projection matrices. Multiple experiments for NLP and vision tasks have been presented that demonstrate reasonable drop in accuracy even for very high (>1000x) compression rates.

---

### Official Review · Reviewer_MH3S · 2022-03-24
**Adopting a classical sketching idea into federated learning applications**

**Rating:** 6
**Confidence:** 4

**Review:**

This paper proposes to use a classical sketching idea in federated SGD-like algorithms to improve the communication efficiency of federated learning algorithms. The proposed method has been used widely in centralized distributed SGD with good efficiency. Thus I am a bit concerned about the novelty of the paper.

Conceptually, I believe the sketching idea can also be adopted on top of federated averaging, i.e., conduct $A^\top A$ over the model updates instead of gradient updates. How does that variant work? Moreover, the error feedback scheme seems to be always helpful for gradient/model compression, does it also help the proposed method?

---

### Official Review · Reviewer_S2Br · 2022-03-24
**Adapting intrinsic gradient compression in federated settings**

**Rating:** 6
**Confidence:** 4

**Review:**

Summary:
A gradient compression technique for federated settings based on the intrinsic dimension concept is proposed. Three variations of the technique are implemented and their tradeoffs in terms of parameter exploration, federation performance and uplink and downlink cost are presented.

Strong and Weak Points:

(S1) Interesting adaptation of intrinsic dimension in federated learning settings for compressing clients' (local) gradients.
(S2) Extensive empirical evaluation against different baselines and on multiple domains.
(S3) Promising insights to employ intrinsic gradient compression techniques against inference attacks.

(W1) Presentation of preliminaries, background, gradient compression and approximation, and algorithms can be improved.

Detailed comments:
(W1) In section 1, it would be better to cite the original intrinsic dimension work the first time it is discussed. In section 2 it would be better to create a table with all the notations you use throughout your work for faster notation indexing. Section 2.2 "the data is averaged", please change to "gradients or weights are averaged". Related Work: Federated Learning, please add some discussion on recent works on weight and gradient pruning in federated settings (e.g., [1], [2]). Please elaborate more on the concept of reconciliation; it is not clear what it is and what its challenges are (maybe pointing to specific lines of the algorithm will be helpful). In your time-varying gradient compression it is not clear why we need twice the bandwidth for downlink and where do the $\theta^{final}$ stems from. For the choice of the compression matrix, why do you need an entire Dxd matrix and not consider the model parameters as a collection of smaller dense matrices? Figures 1 and 2 need to come before table 2 since they are discussed first in the paper. Also you compare against LocalTop-K but you never presented or discussed the technique in the paper.

Please be consistent with your notation, for instance in the static intrinsic gradient compression algorithm why use $\mathcal{L}$ as loss. In section 2.1 wouldn't it be more appropriate to replace $\theta_2$ with $\theta^\prime$; also does it hold that $T < L$? Moreover, why refer to $\ell$ as a task when it has already been defined as loss, maybe another symbol could resolve this. A couple notations and concepts used in algorithm 1 are never presented in the paper (e.g., $A(\sum_{t-1}$) - no need for parenthesis, $z_j$, sketches). In section 3, shouldn't be $A\theta^\prime + \theta_0$ instead of $A\theta^\prime$ in the subscript of function f in the first line of equations? Also, how did you derive the A transpose multiplied with the gradient transpose from the previous line and in equation (5) why is $A\theta_{t+1}^\prime$ equal to $\theta_{t+1}$?


[1] Jiang, Yuang, Shiqiang Wang, Victor Valls, Bong Jun Ko, Wei-Han Lee, Kin K. Leung, and Leandros Tassiulas. "Model pruning enables efficient federated learning on edge devices." arXiv preprint arXiv:1909.12326 (2019).

[2] Bibikar, Sameer, Haris Vikalo, Zhangyang Wang, and Xiaohan Chen. "Federated Dynamic Sparse Training: Computing Less, Communicating Less, Yet Learning Better." arXiv preprint arXiv:2112.09824 (2021).

---

### Decision · Program_Chairs · 2022-03-26

Accept